

# Metabolomic analyses uncover an inhibitory effect of niclosamide on mitochondrial membrane potential in cholangiocarcinoma cells

Thanaporn Kulthawatsiri[1,2], Yingpinyapat Kittirat[1,3], Jutarop Phetcharaburanin[1,2,4], Jittima Tomacha[1], Bundit Promraksa[3], Arporn Wangwiwatsin[1,2,4], Poramate Klanrit[1,2,4], Attapol Titapun[1,2,5], Watcharin Loilome[1,2,4] and Nisana Namwat[1,2,4]

[1] Cholangiocarcinoma Research Institute, Khon Kaen University, Khon Kaen, Khon Kaen, Thailand
[2] Khon Kaen University Phenome Centre, Khon Kaen University, Khon Kaen, Khon Kaen, Thailand
[3] Department of Medical Sciences/Regional Medical Sciences Center 2, Ministry of Public Health, Phitsanulok, Phitsanulok, Thailand
[4] Department of Systems Biosciences and Computational Medicine/Faculty of Medicine, Khon Kaen University, Khon Kaen, Khon Kaen, Thailand
[5] Department of Surgery/Faculty of Medicine, Khon Kaen University, Khon Kaen, Khon Kaen, Thailand

Corresponding author
Nisana Namwat, nisana@kku.ac.th

## ABSTRACT

**Background:** Niclosamide is an oral anthelminthic drug that has been used for treating tapeworm infections. Its mechanism involves the disturbance of mitochondrial membrane potential that in turn inhibits oxidative phosphorylation leading to ATP depletion. To date, niclosamide has been validated as the potent anti-cancer agent against several cancers. However, the molecular mechanisms underlying the effects of niclosamide on the liver fluke *Opisthorchis viverrini* (Ov)-associated cholangiocarcinoma (CCA) cell functions remain to be elucidated. The aims of this study were to investigate the effects of niclosamide on CCA cell proliferation and on metabolic phenoconversion through the alteration of metabolites associated with mitochondrial function in CCA cell lines.

**Materials and Methods:** The inhibitory effect of niclosamide on CCA cells was determined using SRB assay. A mitochondrial membrane potential using tetramethylrhodamine, ethyl ester-mitochondrial membrane potential (TMRE-MMP) assay was conducted. Liquid chromatography-mass spectrometry-based metabolomics was employed to investigate the global metabolic changes upon niclosamide treatment. ATP levels were measured using CellTiter-Glo® luminescent cell viability assay. NAD metabolism was examined by the $NAD^+$/NADH ratio.

**Results:** Niclosamide strongly inhibited CCA cell growth and reduced the MMP of CCA cells. An orthogonal partial-least square regression analysis revealed that the effects of niclosamide on suppressing cell viability and MMP of CCA cells were significantly associated with an increase in niacinamide, a precursor in NAD synthesis that may disrupt the electron transport system leading to suppression of $NAD^+$/NADH ratio and ATP depletion.

**Conclusion:** Our findings unravel the mode of action of niclosamide in the energy depletion that could potentially serve as the promising therapeutic strategy for CCA treatment.

## INTRODUCTION

Cholangiocarcinoma (CCA) is a bile duct cancer described as a malignant tumor developing from the bile duct epithelium. It is mainly caused by *Opisthorchis viverrini* (Ov) infection in continental Southeast Asia. The highest world incidence of CCA has been reported in Northeastern Thailand, especially in Khon Kaen province, where it is 100 times higher than the global incidence (*Alsaleh et al., 2019*). The study of carcinogenesis of CCA has shown that in Thailand CCA is highly associated with the liver fluke (Ov) infection that causes chronic inflammation of the biliary ducts (*Songserm et al., 2009*; *Sripa & Pairojkul, 2008*). Successful treatment of CCA represents a challenging problem because of the difficulty of diagnosing patients at early stages of the disease. Surgery is, thus far, the only potential curative strategy (*Ghouri, Mian & Blechacz, 2015*). At present, the development of new anti-cancer drugs is one of the ultimate goals of cancer research (*Mohan, Thiagarajan & Chandrasekaran, 2014*, *2017*). The simultaneous rise of molecularly targeted treatments and immune checkpoint inhibitors represents a significant development. These new treatments hold the promise of completely transforming how advanced cholangiocarcinoma is treated (*Rizzo & Brandi, 2021*). Currently, *in vitro* investigations have demonstrated that niclosamide can be utilized not only as a standalone treatment but also in combination therapies (*Ren et al., 2022*). The clinical trials using niclosamide for cancer therapy have been explored in many cancers such as colon cancer (*Osada et al., 2011*) and prostate cancer (*Liu et al., 2015*). The procedure for drug development from the initial to the final steps is extremely expensive and time-consuming. Therefore, proving the novel use from existing drugs which are already known for their pharmacokinetics, safety information and have usually been approved for human use, is considered prompter and more economical than testing the newly synthesized drugs.

In 1953, Bayer chemotherapy research laboratory discovered niclosamide, which was developed as a drug to kill snails (*Andrews, Thyssen & Lorke, 1982*). Since 1960, niclosamide has been approved by the U.S. Food and Drug Administration (US FDA) and used as an anti-helminthic drug to treat tapeworm infection in humans. It was marketed as Yomesan (*Chen et al., 2018*). Niclosamide has been reported to act as a mitochondrial uncoupler of the electron transport chain leading to inhibition of oxidative phosphorylation and stimulation of adenosine triphosphatase activity in myeloma cells. Recent studies have demonstrated that niclosamide possesses a strong *in vitro* anticancer activity against a wide range of cancer cells such as colon, breast, prostate, and many other cancer types (*Al-Hadiya, 2005*; *Li et al., 2014*). In addition, niclosamide has the capacity to enhance immunotherapy by regulating pathways such as PD-1/PDL-1 (*Jiang, Li & Ye, 2022*).

To date, the cytotoxic activity of niclosamide against CCA cells has not been reported. Thus, the objectives of this study were to investigate the inhibitory effect of niclosamide on cell proliferation and the mitochondrial membrane potential (MMP) and to elucidate metabolic phenoconversion upon niclosamide treatment in CCA cells using ultra-high performance liquid chromatography quadrupole time-of-flight mass spectrometry (UHPLC-QTOF-MS) metabolomics. Our findings revealed the metabolic phenoconversion linked to niclosamide's cytotoxic effects and its influence on regulating the mitochondrial membrane potential in CCA cells, providing a deeper understanding of the niclosamide mechanism on CCA cell viability.

## MATERIALS AND METHODS

### Reagents

Niclosamide, dimethylsulfoxide (DMSO) and sulforhodamine B were purchased from Sigma-Aldrich (St. Louis, MO, USA). High-performance liquid chromatography (HPLC)-grade methanol, acetone, chloroform, and water were procured from Merck (Darmstadt, DE).

### Cell culture

Human Ov-associated CCA cell lines including poorly differentiated cells; KKU-100, KKU-055 and KKU-452, human mixed papillary with non-papillary cells; KKU-213A, and the highly differentiated immortalized human cholangiocyte cell line; MMNK-1 were obtained from the Japanese Collection of Research Bioresources Cell Bank (Osaka, Japan). The moderately differentiated CCA cells; KKU-213C were established and characterized by Sripa B, as described previously (*Sripa et al., 2020*). Primary Dermal Fibroblast; Normal, Human, Adult (ATCC® PCS-201-012™) were purchased from American Type Culture Collection (ATCC) (Manassas, VA, USA). Cells were cultured in Ham's F-12 nutrient mixture, with the addition of 10% heat-inactivated fetal bovine serum (Thermo Fisher Scientific, Waltham, MA, California, USA) 100 U/mL penicillin and 100 μg/mL streptomycin at 37 °C in a humidified incubator containing 5% $CO_2$.

### Niclosamide cytotoxicity

Niclosamide was dissolved in methanol and acetone at a ratio of 1:1 to obtain a stock solution, which was freshly prepared (*Balgi et al., 2009*). Cells were seeded into 96-well flat-bottom microtiter plates at $2 \times 10^3$ cells/mL and allowed to adhere for 12 h. Cells were incubated in niclosamide at different concentrations (0, 0.2, 0.4, 0.8 and 1.0 μM) for 48 h. Untreated cells were incubated in culture media with 0.1% methanol-acetone. Cell viability was determined by sulforhodamine B (SRB) assay. Doses of niclosamide were selected based on 25%, 50% and 75% inhibitory concentrations ($IC_{25}$, $IC_{50}$ and $IC_{75}$, respectively) for further experiments.

### Cell viability assay

The sulforhodamine B (SRB) assay was used to determine cell viability, as described previously (*Namwat et al., 2008*). After treatment, the cells were fixed with 10%

trichloroacetic acid and 0.4% SRB in 1% acetic acid was added. The protein-bound stain was solubilized with 10 mM Tris base at pH 10.5. Absorbance was measured at 540 nm using a microplate reader (TECAN Trading, Männedorf, CH, Switzerland).

## Mitochondrial membrane potential assay

KKU-100 and KKU-213A cells were plated into 24-well flat-bottom plates at $2 \times 10^4$ cells/mL. The cells were treated with niclosamide for 48 h. Cells were stained using a TMRE-MMP Assay Kit according to the manufacturer's instructions (ab113852, Cambridge, UK) as described previously (*Kittirat et al., 2021*). Then, the cells were visualized by confocal microscopy (Zeiss LSM 800; Carl Zeiss, Jena, Germany).

## ATPase activity assay

CCA cells were plated into 96-well flat-bottom black plates at $2 \times 10^4$ cells/mL. After treatment with niclosamide for 48 h, CellTiter-Glo® Luminescent Cell Viability Assay (Promega, Madison, WI, USA) was used for ATP measurement as described in (*Kittirat et al., 2021*). ATP disodium salt hydrate (Sigma-Aldrich, St. Louis, MO, USA) was used as a standard. The luminescence was determined at 470 nm using SpectraMax® microplate readers (MDS Analytical Technologies, Sunnyvale, CA, USA).

## Sample collection and preparation for UHPLC-QTOF-MS analysis

KKU-100 and KKU-213A cells were seeded at $2 \times 10^5$ cells/mL into 100-mm cell culture dishes. After treatment with niclosamide for 48 h, the treated cells were detached using trypsin-EDTA and $2 \times 10^6$ cells collected for each condition. Then, the collected cells were centrifuged at 358 $g$ at 4 °C for 5 min. The cell pellets were washed with PBS 3 times to remove the remaining medium and immediately stored at −80 °C. For metabolite extraction, the collected cells were resuspended with methanol, ultrasonicated at 40% amplitude for three cycles. The lysed cells were phase extracted using water/methanol/ chloroform (1:1:3 v/v), incubated on ice for 20 min and centrifuged at 1,431 $g$ at 4 °C for 20 min (*Lauri et al., 2016*). The aqueous phase was collected and evaporated by CentriVap® Vacuum Concentrator (Labconco, Kansas City, MO, USA). The dried aqueous extracts were reconstituted in a 200 μL solvent mixture of isopropanol (IPA)/ acetonitrile (ACN)/H$_2$O (2:1:1) and centrifuged for 20 min at 13,000 $g$, 4 °C.
The conditioned media were collected as follows: 50 μl of culture media and 150 μl of isopropanol (IPA) were added into a 1.5 ml tube and mixed well then incubated for 24 h at −20 °C for protein precipitation. Then, the sample tubes were centrifuged at 15,300 $g$ for 10 min at 4 °C. Twenty μL of each sample were taken and pooled in a 1.5 mL tube to make up the quality control (QC) and the 150 μL of sample was transferred to a glass vial insert.

## UHPLC-QTOF-MS metabolomics

Global metabolomics was carried out by the Khon Kaen University Phenome Centre (KKUPC) using ultra-high performance liquid chromatography coupled with electrospray ionization-quadrupole time-of-flight mass spectrometry (compact UHPLC ESI-Q-TOF MS, Bruker, DE). The aqueous phase extracts were analyzed on a reverse-phase UHPLC system as previously described (*Kittirat et al., 2021*; *Poasakate et al., 2021*).

## Data pre-processing and metabolite identification

The data were then imported to MetaboScape software 5.0 software (Bruker, Billerica, MA, USA) for processing. In MetaboScape, the bucket table parameters were generated by using T-ReX 3D (LC-QTOF) workflow. Detection of molecular features was set 1,500 counts of intensity threshold with a minimum peak length of eight spectra. Assignment of metabolites was performed by comparing the MS/MS fragmentation patterns of detected features against the public database, human metabolome database (HMDB), METLIN, Bruker Metabobase and LipidBlast database. The level of assignment (LoA) included (1) accurate mass matched to database indicating tentative assignment, (2) accurate mass matched to database and tandem MS spectrum matched to *in silico* fragmentation pattern, (3) tandem MS spectrum matched to database or literature, (4) retention time and the molecular mass matched to standard compound, and (5) MS/MS spectrum matched to standard compound (*Vorkas et al., 2015*).

## NADH activity assay

The treated cells were collected by scraping for $2 \times 10^6$ cells of each condition. Then, cells were washed with cold PBS, centrifuged at 358 $g$ for 5 min, and the supernatant was discarded. The $NAD^+$/NADH ratio was determined using the NAD/NADH Assay Kit following the manufacturer's instructions (ab65348; Abcam, Cambridge, UK).
The NADH/NAD Extraction Buffer 400 μL was added to the extracted cells by two freeze/thaw cycles (20 min on dry ice followed by 10 min at RT) and vortex for 10 s. After that, the lysed cells were centrifuged at top speed for 5 min at 4 °C. The supernatant was collected and transferred into a new tube. The extracted cells were filtrated to separate the enzymes through a 10 kD Spin Column (ab93349; Abcam, Cambridge, UK) and centrifuged at 10,000 $g$ for 10 min at 4 °C. The 200 μL of extracted samples were added into 1.5 mL tube and heated at 60 °C for 30 min. The samples were mixed with reagents following the manufacturer's instructions (ab65348; Abcam, Cambridge, UK) The reactions were determined at 450 nm using SpectraMax® microplate readers (MDS Analytical Technologies, Sunnyvale, CA, USA).

## Statistical analysis

Data modeling and statistical analysis were conducted using Principle Component Analysis (PCA), Orthogonal Partial Least Squares (O-PLS) for regression analysis, and Orthogonal Partial Least Squares for Discriminant Analysis (O-PLS-DA) in SIMCA software version 14.1 (Umetrics, Umeå, Sweden). Data was scaled using Pareto scaling, and two principal components were accounted to reflect the spectral information. S-plot analysis was utilized to identify the differentiating metabolites among the groups. The statistical analysis of biological results was processed using GraphPad Prism 5.0 software (GraphPad. Software Inc., La Jolla, CA, USA) and performed using ANOVA and Student's t-test. A $p$-value of less than 0.05 was considered statistically significant.

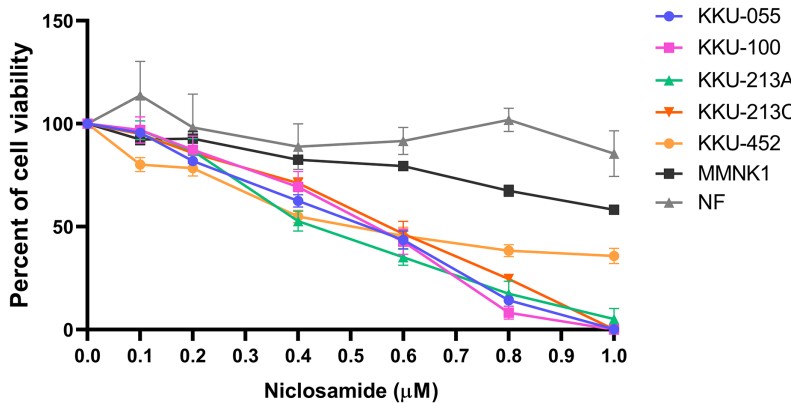

**Figure 1  The inhibitory effect of niclosamide on CCA cell viability.** Primary normal human dermal fibroblasts (NF), human cholangiocyte cells (MMNK-1) and CCA cells (KKU-100, KKU-213A, KKU-213C, KKU-452 and KKU-055) were treated with various concentrations of niclosamide for 48 h. 0.1% (v/v) methanol-acetone was used as a control. Cell viability was determined by SRB assay. The significant difference was determined using two-way ANOVA (*$p < 0.05$, **$p < 0.01$ and ***$p < 0.001$). Error bars represent the standard deviation (SD) of triplicate experiments.

## RESULTS

### Niclosamide inhibited viability of CCA cells

The cytotoxic effect of niclosamide on 5 CCA cell lines, human cholangiocyte cell line (MMNK-1) and primary cultured fibroblasts was determined using an SRB assay at various niclosamide concentrations (0–1.0 μM) for 48 h. We found that niclosamide inhibited CCA cell viability in a dose-dependent manner greater than MMNK-1, whereas niclosamide showed no effects on normal fibroblasts (Fig. 1). The half-maximal inhibitory concentrations ($IC_{50}$) of niclosamide on KKU-100 and KKU-213A, KKU-213C, KKU-055 and KKU-452 were 0.55 ± 0.03 μM and 0.42 ± 0.03 μM, 0.57 ± 0.05 μM, 0.53 ± 0.04 μM and 0.51 ± 0.07 μM respectively. The inhibitory concentrations at 25, 50 and 75 ($IC_{25}$, $IC_{50}$ and $IC_{75}$) of niclosamide on KKU-100, KKU-213A, KKU-213C, KKU-055 and KKU-452 cells are shown in Table 1.

### UHPLC-QTOF-MS metabolomics demonstrated metabolic shifts upon niclosamide treatment in CCA cells

In this study, global metabolic profiling of niclosamide-induced cytotoxicity in CCA cells for 48 h was investigated using UHPLC-QTOF-MS analysis. Spectral data obtained from aqueous extracts of niclosamide-treated CCA cells and their untreated control counterparts were analyzed using multivariate statistical analysis. To maximize the meaningfulness of the metabolome data, pairwise PCA comparison models were also constructed. Niclosamide-treated CCA cells at $IC_{25}$, $IC_{50}$ and $IC_{75}$ were compared with their untreated control in both positive and negative ionization modes of KKU-100 (Fig. S1) and KKU-213A (Fig. S2). The PCA score plot of KKU-100 showed that the niclosamide-treated $IC_{75}$ group was clearly separated from the control group, with 50.9%

**Table 1 The inhibitory concentrations at 25, 50 and 75 values of niclosamide on CCA cells, cholangiocytes and normal fibroblasts.**

| CCA cells | $IC_{25}$ (µM) ± SD | $IC_{50}$ (µM) ± SD | $IC_{75}$ (µM) ± SD |
|---|---|---|---|
| KKU-100 | 0.33 ± 0.06 | 0.55 ± 0.03 | 0.70 ± 0.02 |
| KKU-213A | 0.26 ± 0.02 | 0.42 ± 0.03 | 0.72 ± 0.04 |
| KKU-213C | 0.33 ± 0.04 | 0.57 ± 0.05 | 0.79 ± 0.02 |
| KKU-055 | 0.26 ± 0.02 | 0.53 ± 0.04 | 0.53 ± 0.04 |
| KKU-452 | 0.29 ± 0.15 | 0.51 ± 0.07 | – |
| MMNK-1 | 0.67 ± 0.03 | – | – |
| NF | – | – | – |

of the variance in data explained by the first principal component (PC1) and 28.0% by the second principal component (PC2), and $Q^2$ = 0.54 (Fig. 2A). By contrast, pairwise PCA score plots between niclosamide-treated KKU-213A and the untreated control at $IC_{75}$ demonstrated the invalid models (Fig. 2C). Pairwise O-PLS-DA models were additionally built to visualize the specific metabolic alterations with regards to the niclosamide treatment. The O-PLS-DA analysis of niclosamide-treated KKU-100 ($R^2X$ = 0.749; $Q^2Y$ = 0.925; CV-ANOVA $p$-value = 0.0051) and KKU-213A ($R^2X$ = 0.393; $Q^2Y$ = 0.958; CV-ANOVA $p$-value = 0.1334) with untreated control counterparts demonstrated that the metabolic shift was observed only at $IC_{75}$ (Figs. 2B and 2D) in the positive ionization mode. It is, however, noted that clear metabolic alteration was not observed in either PCA or O-PLS-DA analyses of negative ionization mode data (Figs. S1 and S2).

## Niclosamide suppressed mitochondrial membrane potential of CCA cells

We examined the mitochondrial membrane potential (MMP) of CCA cells following exposure to niclosamide at different inhibitory concentrations ($IC_{25}$, $IC_{50}$, and $IC_{75}$) for 48 h. This assessment was conducted using the TMRE-MMP assay, where TMRE serves as a specific indicator of membrane potential. Carbonyl cyanide-4-(trifluoromethoxy) phenylhydrazone (FCCP) was employed as a negative control. We observed a significant accumulation of TMRE in untreated KKU-100 cells (Fig. 3A) and KKU-213A cells (Fig. 3B), indicated by the red color, whereas its intensity exhibited a dose-dependent reduction in the treated group. Moreover, the TMRE intensities of the treated groups were quantified and compared with the untreated groups. The results showed that niclosamide at $IC_{25}$, $IC_{50}$, and $IC_{75}$ significantly suppressed TMRE intensity in both CCA cell lines. This data shows that niclosamide inhibited MMP in CCA cells.

## O-PLS regression analysis revealed the correlation of niclosamide-suppressed mitochondrial membrane potential and metabolic phenoconversion

To further our understanding of the metabolic phenoconversion associated with niclosamide-suppressed mitochondrial membrane potential, co-analysis between

![PeerJ]

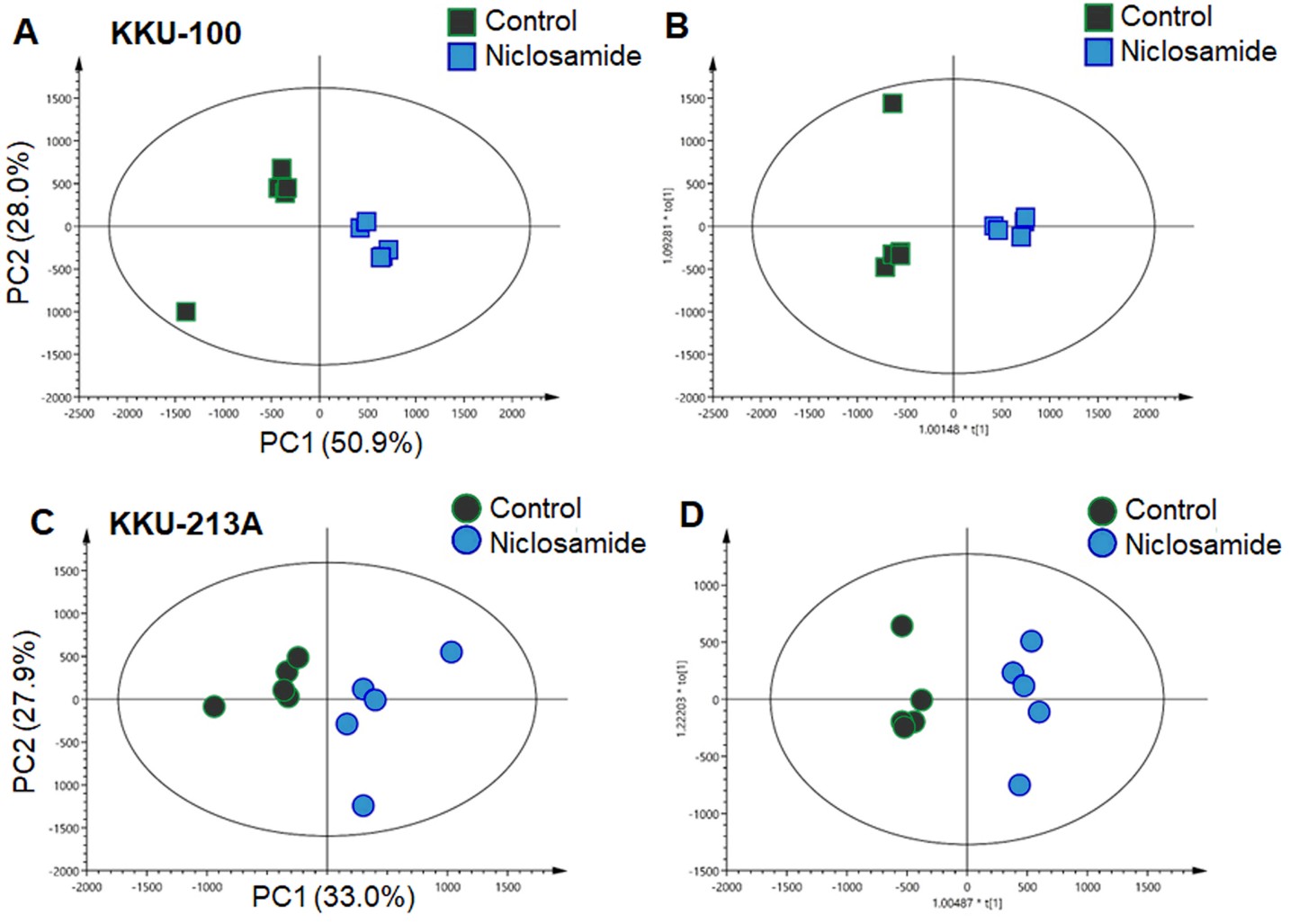

**Figure 2 Multivariate analysis of the metabolic profiles of niclosamide-treated CCA cells.** (A and C) The PCA score plots of the treated and control groups in KKU-100 and KKU-213A cells, respectively. The O-PLS-DA score plots of (C) KKU-100 and (D) KKU-213A cells in control *versus* the treated group models.

metabolome data and TMRE percentage was conducted using O-PLS regression analysis. In positive electrospray ionization mode, both KKU-100 and KKU-213A treated cells contained intracellular metabolic features that were significantly correlated with a lower percentage of TMRE (Fig. 4A). The S-plots derived from the altered metabolic profiles were determined based on the variable importance in projection (VIP) value of greater than 1.0 and p(corr) cut off value of |0.6|. The selected candidate variables are colored in red, representing the metabolites with similarly increased in niclosamide treatment in both CCA cells (Fig. 4B). The results demonstrated that higher intracellular metabolite relative concentrations of niacinamide, *N8*-acetylspermidine and myristamide were positively correlated with a lower percentage of TMRE, hence the association with the inhibition of mitochondrial membrane potential (Fig. 4C, Table 2).

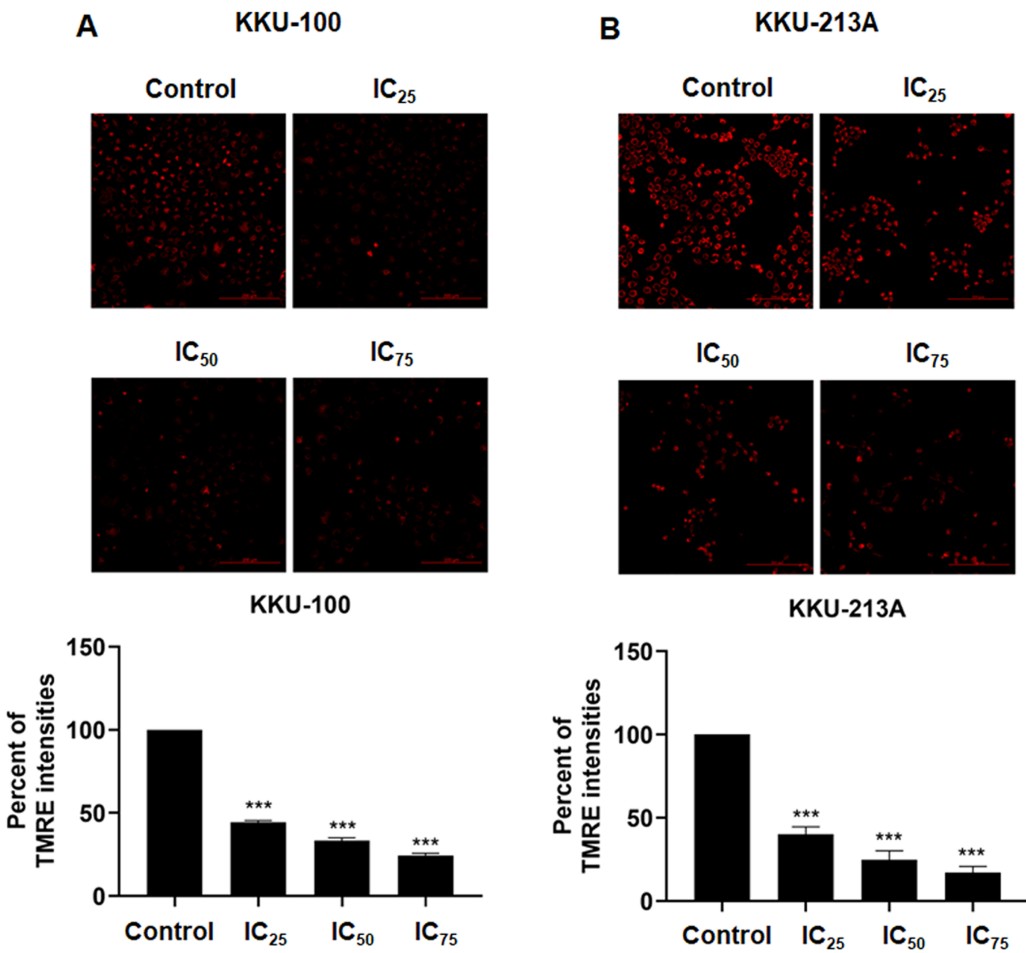

**Figure 3** **The inhibitory effect of niclosamide on mitochondrial membrane potential of CCA cells.**
(A) CCA cells (KKU-100 and KKU213A) were treated with niclosamide for 48 h and incubated with
TMRE. The TMRE staining cells were observed under confocal microscopy. Negative control (NC) cells
were incubated with FCCP before TMRE staining. (B) TMRE intensities of KKU-100 and KKU-213A
were quantified using ImageJ. Error bars represent the standard deviation (SD). The significant difference
was determined using unpaired t-tests (***$p < 0.001$) compared to control group.

## Niclosamide suppressed NAD metabolism and ATP levels of CCA cells

Regarding to the metabolites correlated with the inhibition of mitochondrial membrane
potential, niacinamide was the metabolite with the highest VIP and p(corr) values
(Table 2). To elucidate the increasing of niacinamide level in niclosamide-treated CCA
cells, $NAD^+/NADH$ ratio and ATP level in KKU-100 and KKU-213A were determined
after treatment with niclosamide for 48 h. Our results exhibited that niclosamide
significantly suppressed the $NAD^+/NADH$ ratio (Fig. 5A) and ATP level (Fig. 5B) in a
dose-dependent manner. This result suggests that niclosamide inhibits a redox reaction of
$NAD^+/NADH$ associated with energy production in CCA cells.

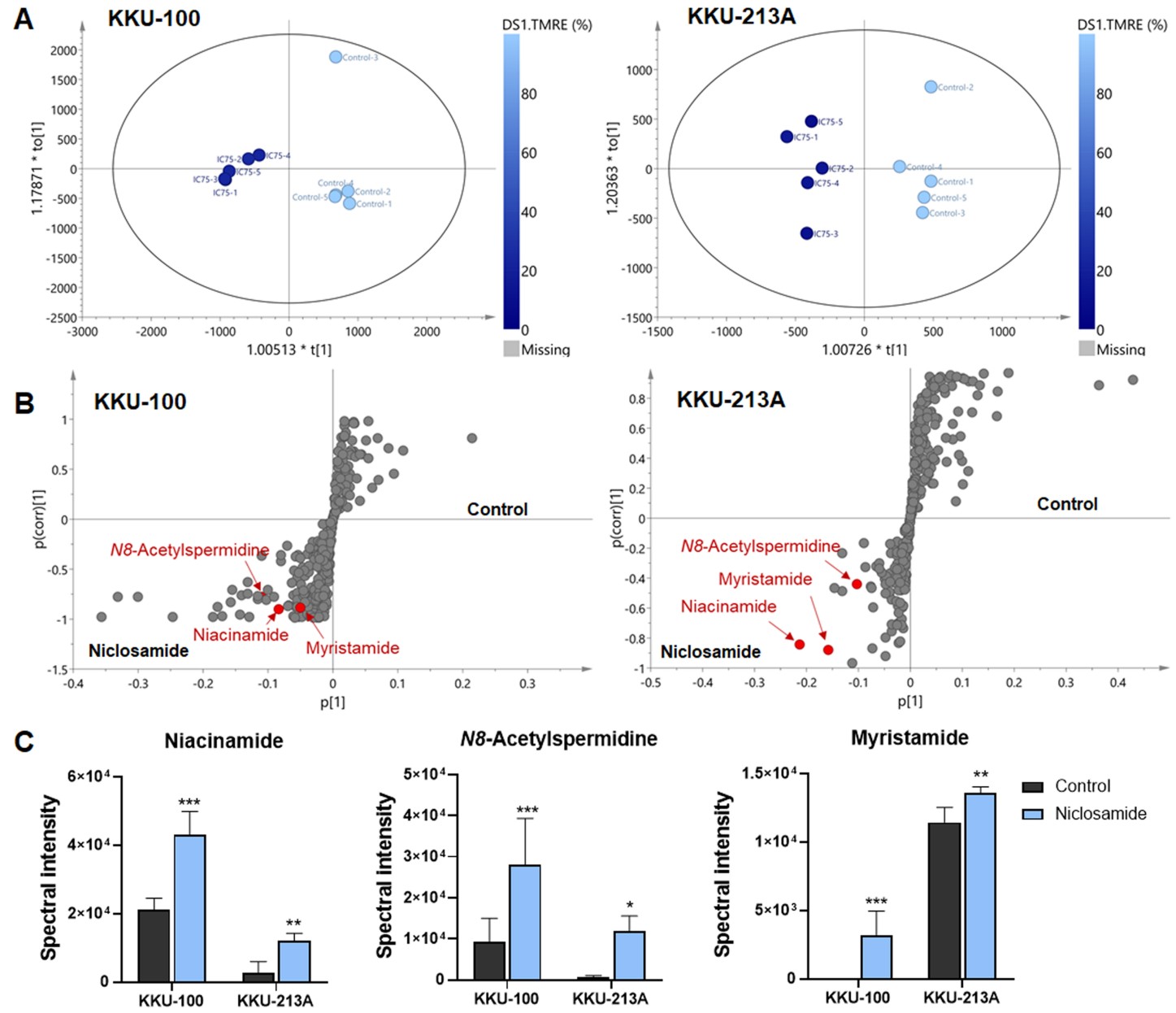

**Figure 4 O-PLS regression analysis of the metabolic profiles with mitochondrial membrane potential in niclosamide-treated CCA cells.** (A) The O-PLS regression score plot is based on TMRE intensities in KKU-100 and KKU-213A cells. (B) The S-plots derived from the altered metabolic profiling of KKU-100 and KKU-213A cells in control *versus* the treatment groups. (C) Spectral intensities of significant metabolites associated with TMRE intensities in KKU-100 and KKU213 treated niclosamide. Error bars represent the standard deviation (SD) of samples ($n = 5$). The significant difference was determined using ANOVA ($^*p < 0.05$, $^{**}p < 0.01$, $^{***}p < 0.001$) compared to control group.

## DISCUSSION

Niclosamide is the U.S. Food and Drug Administration (US FDA)-approved anthelmintic drug for gastrointestinal tapeworm infection. It is currently listed on the World Health Organization's list of essential medicines (*Andrews, Thyssen & Lorke, 1982*).

The mechanism of niclosamide action is an uncoupling of mitochondrial oxidative

**Table 2 The candidate intracellular metabolic identification of KKU-100 and KKU-213 cells associated with mitochondrial membrane potential after treatment with niclosamide.**

| Metabolites | Adduct | m/z | RT | p(corr) | | VIP | | Δppm | LoA |
|---|---|---|---|---|---|---|---|---|---|
| | | | | KKU-100 | KKU-213A | KKU-100 | KKU-213A | | |
| Niacinamide | $[M+H]^+$ | 123.05 | 1.34 | 0.90 | 0.84 | 1.33 | 3.27 | 0 | 2 |
| N8-Acetylspermidine | $[M+H]^+$ | 188.17 | 1.00 | 0.77 | 0.88 | 1.78 | 2.41 | 1 | 3 |
| Myristamide | $[M+H]^+$ | 228.23 | 16.02 | 0.89 | 0.87 | 0.79 | 1.11 | 4 | 3 |

**Note:**
m/z, observed mass-to-charge ratio; RT, retention time (min); LoA, level of Assignment, 2; accurate mass and tandem MS spectrum matched to database or literature, 3; accurate mass similarly to literature.

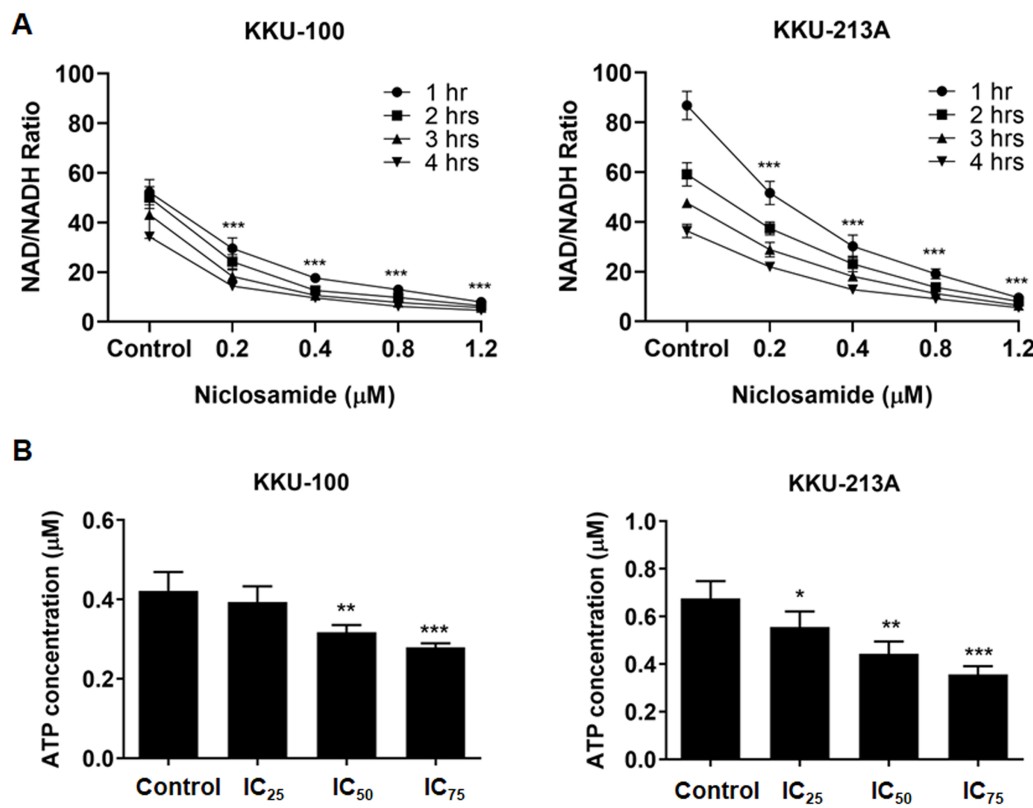

**Figure 5 The inhibitory effect of niclosamide on NAD metabolism and ATP levels in CCA cells.** (A) $NAD^+$/NADH ratio in KKU-100 and KKU-213A cells after various concentrations of niclosamide treatment for 1, 2, 3 and 4 h. (B) ATP levels of KKU-100 and KKU-213A were determined using the CellTiter-Glo® luminescent cell viability assay. The ATP concentrations at 0.01, 0.1 and 1 μM were used to create a standard curve. Error bars represent the standard deviation (SD). The significant difference was determined using ANOVA (*$p < 0.05$, **$p < 0.01$, ***$p < 0.001$) compared to control group.

phosphorylation (*Chen et al., 2018*), contributing to inhibition of ATP synthesis and interruption of energy metabolism (*Al-Hadiya, 2005*). Previous studies reported a strong *in vitro* anti-cancer activity of niclosamide against a wide range of cancer cells such as leukemia, breast cancer, prostate cancer, hepatocellular carcinoma, and glioblastoma by suppressing several signaling pathways in cancer progression (*Hamdoun, Jung & Efferth,*

2017; *Li et al., 2014*). Niclosamide affects CCA cell proliferation in a dose-dependent manner.

The $IC_{50}$ values of niclosamide on treated KKU-100 and KKU-213A, KKU-213C, KKU-055 and KKU-452 cells were in the same range as those of other treated cancer cells (*Li et al., 2014*). The $IC_{50}$ values of gemcitabine, a standard chemotherapeutic drug for CCA, on KKU-213A cells were 100 times higher than that of niclosamide (*Dokduang et al., 2010*). This suggests that niclosamide had high potency against Ov-associated CCA cell proliferation with an $IC_{50}$ value of less than 1 µM. The mode of action of niclosamide is by induction of mitochondrial uncoupling that leads to loss of mitochondrial membrane potential and affects mitochondrial functions (*Frayha et al., 1997*; *Sheth, 1975*; *Mohan, Thiagarajan & Chandrasekaran, 2015*; *Mohan et al., 2016*). In the current study, we demonstrated that the mechanism of niclosamide on mitochondrial uncoupling affects CCA cell growth *in vitro*. Our results showed that niclosamide treatment reduces mitochondrial membrane potential in both CCA cell lines, which interrupts the mitochondrial function of CCA cells resulting in ATP depletion. Thus, our data support the mechanism of the niclosamide action towards mitochondrial membrane potential suppression in CCA cells that is possibly related with oxidative phosphorylation of mitochondria.

This study also investigated the metabolic changes upon niclosamide treatment in CCA cells using LC-MS-based metabolomics. Our metabolomics data demonstrated that the metabolic alteration induced by niclosamide was correlated with the suppression of mitochondrial membrane potential. The altered metabolites in both CCA cell lines, KKU-213A and KKU-100, including niacinamide, N8-acetylspermidine and myristamide, were increased in their relative concentrations upon treatment with niclosamide.

This study employed a metabolomics method to elucidate the mechanism by which niclosamide influences cancer metabolism. Three varieties of metabolites, *N8*-acetylspermidine, myristamide, and niacinamide, were observed to be elevated in CCA cells treated with niclosamide. Normally, *N8*-acetylspermidine does not accumulate in tissues but rather appears to be rapidly deacetylated back to spermidine (*Nayak et al., 2020*). *N8*-acetylspermidine is a key mediator of the ischemic cascade, by regulating apoptosis or programmed cell death. Myristamide is a fatty amide of myristic acid found to accumulate in CCA cells. This fatty acid amide was primarily isolated from harmful algae and identified as a toxic metabolite (*Bertin et al., 2012*). These metabolites are hypothetically toxic metabolic products that may induce cancer cell death.

Strikingly, our data revealed that niacinamide was the most highly correlated metabolite with mitochondrial dysfunction in niclosamide-treated CCA cells. Niacinamide, also known as nicotinamide or vitamin B3, is the main precursor of nicotinamide adenine dinucleotide ($NAD^+$), an essential coenzyme for DNA repair, glycolysis, and oxidative phosphorylation (*Tan et al., 2019*). This is consistent with our findings that niclosamide suppresses the $NAD^+$/NADH ratio and ATP levels in CCA cells. The $NAD^+$/NADH ratio is involved in central carbon metabolism, nucleotide synthesis, lipid metabolism, as well as amino acid metabolism (*Luengo et al., 2021*). In this regard, our results indicate that the increase of niacinamide could be due to incomplete NAD synthesis, resulting in a decrease

of NAD that affects energy metabolism since the role of NAD is as an electron shuttle during cellular respiration, which is important in ATP synthesis. This suggests that niclosamide could inhibit the redox reaction of $NAD^+/NADH$ associated with metabolic reactions contributing to the induction of CCA cell death. For these reasons, cancer cells lack energy, potentially inhibiting cancer cell proliferation and leading to cancer cell death.

Collectively, this study, for the first time, indicates that niclosamide inhibits CCA cell proliferation by accumulating the precursor of NAD leading to suppress mitochondrial membrane potential, which in turn interrupts the mitochondrial function resulting in depletion of NAD and ATP levels. Altering metabolites from UHPLC-QTOF-MS metabolomics analysis demonstrates the significant alteration of metabolites related to cancer cell function upon niclosamide treatment. Our findings have further strengthened the concept that niclosamide should be considered an anticancer agent. Further exploration is recommended to investigate the potential of combining niclosamide with existing therapies for CCA. Our study limits *in vitro*; therefore, further investigation is required for *in vivo* studies. In a clinical context, the major impediment for niclosamide is poor oral bioavailability, potentially limiting its use as an anticancer drug (*Jiang, Li & Ye, 2022*). Even though preclinical models have produced encouraging data, there is still a need for actual evidence regarding both effectiveness and safety. Therefore, initiatives have been pursued to enhance its bioavailability including reformulating niclosamide for better delivery and stability, modifying the structure of niclosamide to generate derivatives with enhanced efficiency or pharmacokinetics.

## ACKNOWLEDGEMENTS

We would like to acknowledge Prof. Trevor N. Petney for editing the article via the Publication Clinic KKU, Thailand.

### Funding

This work was supported by a grant from Faculty of Medicine, Khon Kaen University (grant. no. IN63234) to Thanaporn Kulthawatsiri and the NSRF under the Basic Research Fund of Khon Kaen University through Cholangiocarcinoma Research Institute to Nisana Namwat. The funders had no role in study design, data collection and analysis, decision to publish, or preparation of the manuscript.

### Grant Disclosures

The following grant information was disclosed by the authors:
Khon Kaen University: IN63234.
Cholangiocarcinoma Research Institute to Nisana Namwat.

### Competing Interests

The authors declare that they have no competing interests.

## Author Contributions

- Thanaporn Kulthawatsiri conceived and designed the experiments, performed the experiments, analyzed the data, prepared figures and/or tables, authored or reviewed drafts of the article, and approved the final draft.
- Yingpinyapat Kittirat performed the experiments, analyzed the data, prepared figures and/or tables, authored or reviewed drafts of the article, and approved the final draft.
- Jutarop Phetcharaburanin analyzed the data, authored or reviewed drafts of the article, and approved the final draft.
- Jittima Tomacha performed the experiments, prepared figures and/or tables, and approved the final draft.
- Bundit Promraksa analyzed the data, prepared figures and/or tables, authored or reviewed drafts of the article, and approved the final draft.
- Arporn Wangwiwatsin conceived and designed the experiments, authored or reviewed drafts of the article, and approved the final draft.
- Poramate Klanrit conceived and designed the experiments, authored or reviewed drafts of the article, and approved the final draft.
- Attapol Titapun conceived and designed the experiments, authored or reviewed drafts of the article, and approved the final draft.
- Watcharin Loilome conceived and designed the experiments, authored or reviewed drafts of the article, and approved the final draft.
- Nisana Namwat conceived and designed the experiments, prepared figures and/or tables, authored or reviewed drafts of the article, and approved the final draft.

## Data Availability

The raw data are available in the Supplemental File.

## Supplemental Information

Supplemental information for this article can be found online at http://dx.doi.org/10.7717/peerj.16512#supplemental-information.

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
