# Peer review of "Metabolomic analyses uncover an inhibitory effect of niclosamide on mitochondrial membrane potential in cholangiocarcinoma cells"

_PeerJ, doi:10.7717/peerj.16512_

## Round 0.1 · original submission · Major Revisions

Reviewer 1 has suggested that you cite specific references. You are welcome to add it/them if you believe they are relevant. However, you are not required to include these citations, and if you do not include them, this will not influence my decision.

I am looking forward to receiving the revised version of your manuscript.


Prof. Yoshinori Marunaka, M.D., Ph.D.

Reviewer 1 ·

Basic reporting

Dear Editor, thank you so much for inviting me to revise this manuscript about CCA.

This study addresses a current topic.

The manuscript is quite well written and organized. English could be improved.
Figures and tables are comprehensive and clear.
The introduction explains in a clear and coherent manner the background of this study.

We suggest the following modifications:
• Introduction section: although the authors correctly included important papers in this setting, we believe the evolving systemic treatment scenario for CCA should be further discussed and some recently published papers added within the introduction ( PMID: 33645367 ; PMID: 33592561; PMID: 36633661; PMID: 32806956 ), only for a matter of consistency. We think it might be useful to introduce the topic of this interesting study.
• Methods and Statistical Analysis: nothing to add.
• Discussion section: Very interesting and timely discussion. Of note, the authors should expand the Discussion section, including a more personal perspective to reflect on. For example, they could answer the following questions – in order to facilitate the understanding of this complex topic to readers: what potential does this study hold? What are the knowledge gaps and how do researchers tackle them? How do you see this area unfolding in the next 5 years? We think it would be extremely interesting for the readers.

However, we think the authors should be acknowledged for their work. In fact, they correctly addressed an important topic, the methods sound good and their discussion is well balanced.

One additional little flaw: the authors could better explain the limitations of their work, in the last part of the Discussion.

We believe this article is suitable for publication in the journal although major revisions are needed. The main strengths of this paper are that it addresses an interesting and very timely question and provides a clear answer, with some limitations.

We suggest a linguistic revision and the addition of some references for a matter of consistency. Moreover, the authors should better clarify some points.

Experimental design

.

Validity of the findings

.

Reviewer 2 ·

Basic reporting

In this study, Kulthawatsiri et al. seek the anti-cancer effects of niclosamide using CCA cells. The topic is novel, but there are a few basic but critical issues to be fixed in this study.
• CCA is very heterogeneous and there are multiple CCA cell lines are available. In previous studies using CCA cell lines, results are inconsistent or controversial depending on CCA cell lines used in experiments. Therefore, multiple (up to 5) CCA cell lines should be included for in vitro studies, using cell lines derived from intrahepatic, perihilar, and extrahepatic CCA. In addition, since CCA cells are very heterogeneous and can be altered after passage, cell authentication is suggested to prove that CCA cells used in the study are authenticated and free of contamination, modification, or transformation. There are several commercially available cell authentication service (such as ATCC), and there are databases for references of STR profiles (such as cellosaurus.org). Some journals do not accept studies of CCA cell lines without authentication, so I think this should be included because this study is based only on in vitro studies using CCA cell lines.

Experimental design

• The control is normal human dermal fibroblasts in this study (Figuere1), but this is not suitable. CCA is a biliary cancer, so the control cells should be normal human bile duct epithelial cells or cholangiocytes. There are commercially available normal human cholangiocytes.
• Control cells should be included in Figure 2-5 to show that niclosamide does not significantly damage normal biliary cells.

Validity of the findings

• Statistical analysis, such as two-way ANOVA and post-hoc analysis should be performed for Figure 1.

·

Basic reporting

The aim and objectives were clear, english, references, M& M used is appropriate

Experimental design

Design of experiment with appropriate methodologies.
Line No. 95 Niclosamide cytotoxicity - is this method is your’s??? if it is referred Citation ???
Line No. 104 Cell viability assay - is this method is your’s??? if it is referred Citation ???
Line No. 111 Mitochondrial membrane potential assay - is this method is your’s??? if it is referred Citation ???
Line No. 117 ATPase activity assay - is this method is your’s??? if it is referred Citation ???
Line No. 124 Sample collection and preparation for UHPLC-QTOF-MS analysis- is this method is your’s??? if it is referred Citation ???
Line No. 162 NADH activity assay - s this method is your’s??? if it is referred Citation ???

Validity of the findings

Results and discussion is good, however, I woudl like to suggest to strengthen discussion part

Additional comments

Line No. 285 – is it J.Bertin et al. 2012 or Bertin et al. 2012

---

## Round 0.2 · Minor Revisions

Dear Dr. Namwat,

I would suggest revising your manuscript according to the remaining comments from Reviewer 3

Yours,

Yoshi

Prof. Yoshinori Marunaka, M.D., Ph.D.

Reviewer 1 ·

Basic reporting

acceptance

Experimental design

acceptance

Validity of the findings

acceptance

Additional comments

acceptance

Reviewer 2 ·

Basic reporting

no comment

Experimental design

no comment

Validity of the findings

no comment

Additional comments

The authors addressed all my comments and I have no further comments.

·

Basic reporting

Authors aims and objecives were substaniated with apporiate literature. How aever I suggest to add the followiing referenecs in introduction and discussion part
Introduction & Discussion part I suggested to use the following references
1. Shalini, M., Kalaivani, T. Rajasekaran, C. and Arul, J. (2014). In vitro protection of biological macromolecules against oxidative stress and in vivo toxicity evaluation of Acacia nilotica (L.) and ethyl gallate in rats. BMC Complementary and Alternative Medicine, 14:257.
2. Shalini, M., Kalaivani, T. and Rajasekaran, C. (2015). In vitro evaluation of antiproliferative effect of ethyl gallate against human oral squamous carcinoma cell line KB. Natural Product Research 29(4): 366-369. http://dx.doi.org/10.1080/14786419.2014.942303.
3. Shalini, M., Kalaivani, T., Balaji, S., Gurung, V., Barpande, M., Agarwal S. and Rajasekaran, C. (2016). Alleviation of 4-nitroquinoline 1-oxide induced oxidative stress by Oroxylum indicum (L.) leaf extract in albino Wistar rats. BMC Complementary and Alternative Medicine, 16 (1): 229. DOI: 10.1186/s12906-016-1186-x
4. Shalini, M., Kalaivani, T. and Rajasekaran, C. (2017). Evaluation of ethyl gallate for its antioxidant and anticancer property against chemical induced tongue carcinogenesis in mice. Biochemical Journal 474: 3011–3025. DOI: 10.1042/BCJ20170316.
Results
Figure 1 - X axis should be concealed with 1.0 µM & Y axis should be concealed with 120%
Either keep Figure 1 or Table 1.

Experimental design

Acceptable methods chosen

Validity of the findings

Acceptable, appriciating the authors efforts

Additional comments

With minor revison the manuscript can be accepted

---

## Round 0.3 · accepted · Accept

Dear Dr. Namwat,

Congratulations!

Yours,

Yoshi

Prof. Yoshinori Marunaka, M.D., Ph.D.

·

Basic reporting

The authrors revised the manuscript and given their updates is appriciable

Experimental design

Fine

Validity of the findings

Done

Additional comments

Appriciate the efforts of the authors, the present form of the manucript can be accepted